# Cannabinoids: Potential for Modulation and Enhancement When Combined with Vitamin B12 in Case of Neurodegenerative Disorders

**DOI:** 10.3390/ph17060813

**Published:** 2024-06-20

**Authors:** Anna Aleksandra Kaszyńska

**Affiliations:** The Centre of Neurocognitive Research, Institute of Psychology, SWPS University of Social Sciences and Humanities, Chodakowska 19/31, 03-815 Warszawa, Poland; akaszynska@st.swps.edu.pl

**Keywords:** cannabinoids, CBD, neurodegeneration, vitamin B12, mitochondrial dysfunction, aggregation

## Abstract

The enduring relationship between humanity and the cannabis plant has witnessed significant transformations, particularly with the widespread legalization of medical cannabis. This has led to the recognition of diverse pharmacological formulations of medical cannabis, containing 545 identified natural compounds, including 144 phytocannabinoids like Δ9-THC and CBD. Cannabinoids exert distinct regulatory effects on physiological processes, prompting their investigation in neurodegenerative diseases. Recent research highlights their potential in modulating protein aggregation and mitochondrial dysfunction, crucial factors in conditions such as Alzheimer’s Disease, multiple sclerosis, or Parkinson’s disease. The discussion emphasizes the importance of maintaining homeodynamics in neurodegenerative disorders and explores innovative therapeutic approaches such as nanoparticles and RNA aptamers. Moreover, cannabinoids, particularly CBD, demonstrate anti-inflammatory effects through the modulation of microglial activity, offering multifaceted neuroprotection including mitigating aggregation. Additionally, the potential integration of cannabinoids with vitamin B12 presents a holistic framework for addressing neurodegeneration, considering their roles in homeodynamics and nervous system functioning including the hippocampal neurogenesis. The potential synergistic therapeutic benefits of combining CBD with vitamin B12 underscore a promising avenue for advancing treatment strategies in neurodegenerative diseases. However, further research is imperative to fully elucidate their effects and potential applications, emphasizing the dynamic nature of this field and its potential to reshape neurodegenerative disease treatment paradigms.

## 1. Introduction

The long-standing relationship between humans and the cannabis plant has evolved through shifting societal views and the legalization of medical cannabis globally. “Medical cannabis” refers to various formulations from the cannabis plant, home to 545 compounds and 144 cannabinoids, like Δ9-THC (Δ9-tetrahydrocannabinol) and CBD (cannabidiol). These cannabinoids are divided into endocannabinoids, phytocannabinoids, and synthetic cannabinoids, with distinct physiological regulatory effects. Vaporization has been recognized as a safer consumption method than traditional combustion. Although the therapeutic potential is evident, uncertainties remain about cannabis’s effect on the immune system. This highlights the need for comprehensive research and knowledge dissemination regarding its immunomodulatory properties, especially as cannabis becomes a common palliative treatment [1]. In the realm of neurodegenerative diseases, recent research examines the potential roles of cannabinoids in maintaining organic homeodynamics. The endocannabinoid system (ECS) appears to play a vital role in neurodegenerative disease development, with cannabinoids potentially normalizing critical homeodynamic properties. Their therapeutic potential, similar to the benefits of omega-3 in slowing amyotrophic lateral sclerosis (ALS), is noteworthy due to cannabinoids’ complexity, which is visible in the regularities of improvement under omega-3 fatty acids’ [2] impact as well as dependency on vitamin B12 deficiency [3]. Advances include β-caryophyllene nanoparticles targeting cannabinoid receptor 2 (CB_2_R) [4] and a micellar system for cannabidiol delivery with high encapsulation efficiency [5]. Maintaining homeodynamics is essential, as disturbances can lead to disorders, contributing to neurotoxicity and inflammation linked with lipid metabolism impairment that might be involved in alterations associated with neurodegenerative disorders, highlighting the hypothesis of their being secondarily a lipid-metabolism-associated disorder. Study [6] suggests an interplay of autophagy, calcium homeostasis, mitochondrial function, and apoptosis with altered lipid metabolism in the Drosophila model of Huntington’s Disease (HD). Research on heterozygous and homozygous zQ175 mice at 12 months of age shows that modulating the mGluR5 receptor can decrease huntingtin protein aggregates, suggesting the therapeutic promise of cannabinoids in diseases like HD [7]. This review investigates the ECS, homeodynamics, and the synergistic effects of cannabinoids and vitamin B12 on neurodegenerative diseases, such as Parkinson’s disease (PD), multiple sclerosis (MS), Alzheimer’s Disease (AD), HD, and ALS. It considers their combined potential to modulate cellular energy, oxidative, and inflammatory responses, impacting neural signal transmission. The narrative emphasizes a holistic approach to studying cellular challenges in neurodegenerative disorders. After discussing the ECS and its role in protein aggregation and microglial function, this article turns to the effects of vitamin B12 on these processes and its combination with cannabinoids for potential therapeutic interventions. Since vitamin B12 seems to be a vital cofactor in numerous cellular processes, it highlights potential therapeutic interventions [8,9,10,11]—the vitamin B complex suppresses neuroinflammation in activated microglia. Thus, this review article focuses on the neurodegenerative diseases regarding, e.g., protein aggregation or mitochondrial dysfunction, ECS, homeodynamics, and the synergistic impact of cannabinoids derived from omega fatty acids and therapeutic potential of their combination with vitamin B12 since both display properties of modulating inflammatory responses. This interconnected narrative underscores the holistic approach needed to address the complex investigation of cellular challenges in neurodegenerative disorders. After describing ECS and its receptors and cannabinoids as well as the relation between aggregation and microglial dysfunction, the focus will be put on Section 3, indicating that it can be hypothesized, given the enhancing effects of CBD observed alongside vitamin B12 in the treatment of MS, that a combination of CBD with vitamin B12 might prove beneficial for neurodegeneration improvement. Particularly when this combination is incorporated within a micellar system, as proposed by Yordanov et al. (2022) [5], it aims at increasing the therapeutic potential.

## 2. Research Main Overview: The Endocannabinoid System with Cannabinoid-Related Orphan GPCRs, Signal Transduction and Cannabinoids

### 2.1. The Endocannabinoid System and Cannabinoid-Related Orphan GPCRs

#### 2.1.1. The Endocannabinoid System

The human ECS is integral to regulating physiology and has become a prime focus for drug discovery. Detailed knowledge of cannabinoid receptor (CBR) structure is critical for understanding the modulation of downstream signaling within the cannabinoid system, positioning the ECS as a significant research subject. The ECS, a complex network, upholds physiological equilibrium and responds to stressors, immune challenges, and other regulatory demands. It encompasses cannabinoid receptors CB1 (CB_1_R) and CB_2_R, endogenous cannabinoids, and enzymes for synthesizing and degrading endocannabinoids, hence influencing diverse biological functions. CB_1_R is mainly in the central nervous system (CNS) and peripheries like the liver and immune cells, modulating neurotransmission. Conversely, CB_2_R is in immune and CNS cells, particularly astrocytes and microglial cells, playing a part in immune response and tissue homeostasis. Anandamide (AEA) and 2-arachidonoylglycerin (2-AG), derived from arachidonic acid, are principal signaling molecules within the ECS. AEA is a partial agonist, and 2-AG is a full agonist at CB_1_R and a CB_2_R agonist. Study [12] reveals the ECS’s role in cataleptogenic processes in HD, presenting cannabinoid-based intervention opportunities—Köfalvi et al. (2020) explored the interaction of A_2_A (adenosine), CB_1_R, and D_1_ receptors in HD, revealing the ECS’s potential involvement in cataleptogenic processes. This suggests promising avenues for cannabinoid-based interventions, particularly in normalizing connexin 43 and addressing neurotoxicity and neuroinflammation. Review made by Campos et al. (2016) [13] underscores ECS’s neuroprotective traits, deeming it a potential neuropsychiatric intervention target. Reviews [14,15,16,17,18] discuss its roles in metabolic regulation, stress, and emotional behavior, indicating therapeutic possibilities for stress-related conditions. Insights into neuroprotection, anti-inflammation, metabolic regulation, and stress response highlight the ECS’s potential as a therapeutic target across various medical conditions. Further research in this field holds promise for developing targeted interventions harnessing the regulatory power ECS. At the core of ECS lie endocannabinoids, signaling molecules derived from essential fatty acids, particularly emphasizing the role of omega-6 fatty acids. One of the primary endocannabinoids, AEA, undergoes synthesis from the omega-6 fatty acid arachidonic acid. Omega-6 fatty acids, particularly arachidonic acid, are vital for synthesizing AEA, a significant endocannabinoid. AEA, synthesized on demand, is an endogenous ligand primarily for CB_1_R, influencing mood, pain perception, and appetite [19,20]. Similarly, 2-AG, also derived from omega-6, acts as a partial agonist at CB_1_R and CB_2_R, affecting immune response and neuroinflammation [21]. As such, dietary omega-6 plays a crucial role in ECS function, with a balanced omega-3 to omega-6 ratio being imperative for optimal health—moreover, it is worth mentioning a review indicating ECB’s or GPCRs’ role in homeodynamics (including inhibition endocannabinoid degradation, uptake, and intracellular transport as a strategic treatment approach) or signal transduction, supporting wellbeing as well as highlighting cannabinoid receptors’ potential in drug development (through use of nanoprecision tools) by underlying their molecular mechanisms of action [22]. The synergistic relationship between omega-3 and omega-6 fatty acids is essential for endocannabinoid synthesis and function, potentially affecting neuroprotection and anti-inflammatory responses [23]. The connection between dietary lipids and the ECS illustrates the importance of the dietary origin of endocannabinoids in physiological regulation, as seen in ALS outcomes. Alpha-linolenic acid (ALA), an unsaturated omega-3 fatty acid from plants, has been associated with reduced mortality risk and slower functional decline in patients with ALS over 18 months [2]. These findings underscore the broader roles of omega-derived endocannabinoids in maintaining health and homeostasis.

#### 2.1.2. Cannabinoid-Related Orphan GPCRs

Orphan G protein-coupled receptors (GPCRs) are identified via genetic sequencing but lack known specific ligands or functions. These cell surface receptors transmit external signals internally and remain ‘orphans’ until their biological roles are defined. GPCRs, structurally similar to CBRs, do not bind classical cannabinoid ligands like Δ9-THC or CBD. Initially deemed orphan receptors due to unidentified endogenous ligands, they are now implicated in pain, inflammation, metabolism, and neurotransmission. Ongoing research is unraveling their functions and potential ligands. Several receptors, such as GPR18, GPR55, transient receptor potential vanilloid 1 (TRPV1), peroxisome proliferator-activated receptors (PPARs), and GPR119, align with the ECS in functionality or cannabinoid interaction [24]. *N*-Methyl-d-Aspartate (NMDA) receptors, crucial for synaptic plasticity and cognitive processes, exhibit cannabinoid interactions that affect neuronal function. NMDA receptor dysregulation is linked to neurodegenerative disease pathologies, suggesting that modulating these receptors may provide neuroprotective therapies thanks to observed properties of metabolic balance maintenance [25]. TRPA1 is involved in pain, inflammation, and chemosensing, with activation by cannabinoids affecting pain and inflammation. GPR55, widely expressed in the CNS and immune system, participates in diverse physiological roles such as pain, bone development, and cancer. Its activation by Δ9-THC and AEA remains controversial in classifying it as a cannabinoid receptor [26]. Evidence of dimerization between CB_2_R and GPR55, and their cross-signaling effects, has emerged, with implications for signaling pathways like Nuclear Factor of Activated T cells (NFAT) and Serum Response Element (SRE) [27]. GPR55 is gaining interest for its role in neurodegenerative diseases, including AD, PD, and ALS, where it may influence amyloid aggregation, tau phosphorylation, dopamine signaling, neuroinflammation, and oxidative stress [28,29,30,31]. GPR55’s activation modulates neurodegenerative processes, suggesting it as a promising therapeutic target. Its involvement in AD relates to Aβ peptide accumulation and tau phosphorylation. In PD, particularly in mitochondrial dysfunction models, GPR55 may offer neuroprotection. Its activation affects dopamine release and motor behavior, as observed in PD animal models. GPR55 is also implicated in immune response regulation in MS, affecting immune cell behavior and blood–brain barrier (BBB) integrity. Understanding GPR55’s roles could open new therapeutic strategies for neurodegenerative diseases, underpinning the importance of continued research in this area.

GPR18, a member of GPCRs’ family, is also known as the *N*-arachidonyl glycine (NAGly) receptor or abnormal-cannabidiol (abn-CBD) receptor. Its activation modulates cellular signaling, including the Gαi/o protein-mediated inhibition of adenylyl cyclase and ERK1/2 MAPK pathway activation. Although GPR18 is related to ECS, its specific functions and ligands are not fully defined. Review perspectives [32,33] suggest GPR18’s involvement in immune responses and inflammation. It is activated by NAGly, a lipid closely related to cannabinoids. Notably expressed in immune cells like macrophages and dendritic cells, GPR18 may play a part in immune regulation. It shows variable expression across tissues: absent in the amygdala, frontal cortex, hippocampus, liver, and muscle; low in the cortex and various other organs [32]; moderate in the lungs and reproductive organs; and strong in the hypothalamus, thyroid, peripheral blood leukocytes, and several brain regions. In the testis, GPR18 mRNA is prevalent in differentiating gametes, particularly mature ones. GPR18 activation influences immune cell behavior and cytokine and chemokine production and appears to play a part in immune surveillance. Microglia, which express GPR18 extensively, respond to NAGly—mimicking abn-CBD and stimulating microglial migration via MAPK in CNS injury contexts. GPR18’s presence in brain areas associated with pain and mood suggests its roles in nociception and anxiety-like behaviors, potentially affecting neurotransmitter release and neuronal excitability. Given that dysregulated neuroinflammation is common in neurodegenerative diseases like AD, PD, or MS, GPR18’s impact on these processes may contribute to the pathology of such conditions [34]. Additionally, GPR119 is found mainly in pancreatic β-cells and the gastrointestinal tract, implicated in glucose regulation and incretin hormone secretion, possibly affecting endocannabinoid levels and cannabinoid receptor signaling. GPR3, GPR6, and GPR12, which structurally resemble CB_1_R and CB_2_R, have not yet had their roles within the ECS fully elucidated, although they may interact with cannabinoids and influence cannabinoid receptor signaling.

Transient receptor potential vanilloid 1 (TRPV1) [35] is a non-selective cation channel belonging to the transient receptor potential (TRP) ion channel family, with a broad presence in the central and peripheral nervous systems, as well as peripheral organs such as the skin, bladder, and gastrointestinal tract. It is best known for mediating thermal nociception and pain but also governs other sensory responses to chemicals and mechanical stimuli. TRPV1 can be activated by several endogenous and exogenous ligands, including specific cannabinoids like AEA and N-arachidonoyl dopamine (NADA). This activation impacts pain perception, inflammation, and other physiological functions [35]. It also interacts with the ECS components, including CBRs, to influence cellular signaling pathways. While TRPV1 agonists are explored for their potential in pain management and reducing inflammation, their application is challenged by side effects such as the induction of a burning pain sensation. Nevertheless, TRPV1’s role in modulating pain and neuroinflammatory responses is significant. In neurodegenerative conditions such as AD and MS, TRPV1 may influence neuropathic pain and neuroinflammation, and it may enhance Aβ clearance in glial cells through autophagy [36]. Further, there is growing evidence that TRPV1 activation might offer neuroprotection by playing a role in synaptic plasticity and defending against excitotoxicity, underscoring its potential as a therapeutic target.

Peroxisome proliferator-activated receptors (PPARs) are nuclear receptors that act as transcription factors, directing gene expression involved in numerous metabolic pathways. These receptors are found throughout the body, including in the brain, liver, adipose tissue, and immune cells, and are essential for lipid metabolism, glucose homeostasis, inflammation, and cell differentiation. PPARs also interface with ECS. Both PPARα and PPARγ are known to interact with endocannabinoids [37] such as AEA and 2-AG, as well as plant-derived cannabinoids including Δ9-THC and CBD. These interactions can lead to the PPAR-mediated modulation of gene expression, influencing metabolism, inflammation, and other vital physiological activities. The therapeutic exploration of PPAR activation is wide-ranging, covering metabolic disorders like diabetes and dyslipidemia, inflammatory conditions, neurodegenerative diseases, and cancer. Neurodegenerative diseases, in particular, characterized by inflammation, oxidative stress, and disrupted lipid metabolism, present a significant opportunity for PPAR-targeted therapies. Activating PPARs has shown promise in AD [38], potentially enhancing neuronal and myelin maturation, and mitochondrial function, while reducing neuroinflammation and toxicity. Moreover, their role in other diseases like PD and MS is also being investigated, with PPARs offering a potential avenue for controlling neuroinflammation and supporting neuronal survival [39,40].

Adenosine receptors, a subset of GPCRs, mediate the various physiological effects of the neurotransmitter adenosine. They are involved in modulating cannabinoid receptor signaling and the impact of cannabinoids on neurotransmission, inflammation, and other biological processes. Notably, A_2_A receptors have a significant role in modulating neuroinflammation and offering neuroprotection. In PD, antagonists of A_2_A receptors are being researched as potential therapeutic agents for their potential to mitigate neuroinflammation and bolster dopaminergic neurotransmission. Serotonin receptors, specifically the 5-HT1A, 5-HT2A, and 5-HT3 subtypes, also demonstrate interactions with cannabinoids, affecting mood, anxiety, and neurological functions. These receptors are known for their involvement in mood regulation and cognitive processes that are often impaired in neurodegenerative diseases such as AD and PD. Alterations in serotonin signaling may contribute to the psychiatric and cognitive symptoms observed in these disorders.

### 2.2. Signal Transduction

Signal transduction mechanisms, involving guanosine triphosphate (GTP), cyclic adenosine monophosphate (cAMP), and β-arrestin, play essential roles in the actions via opioid receptors [41,42]. GPCR mu opioid receptor signaling at the plasma membrane mediates analgesia via Gαi, while endosomal ERK signaling contributes to maladaptive tolerance, respiratory depression, and gastrointestinal dysmotility, leading to constipation. Current research highlights GPCR signaling from endosomes, the Golgi apparatus, and nuclear membrane sites as significant and novel therapeutic targets for various disorders [41]. CB_1_R and CB_2_R are part of the GPCR family. The activation of these receptors by cannabinoids triggers a series of signaling events that are central to a wide array of physiological functions.

The GTP signaling pathway plays a critical role in cannabinoid receptor function [43,44]. The activation of CBRs by cannabinoids induces conformational changes that activate heterotrimeric G proteins composed of α, β, and γ subunits. The Gα subunit, which binds GDP in its inactive state, replaces GDP with GTP in response to cannabinoid binding, thereby activating the receptor. This exchange prompts the Gα subunit to dissociate from the Gβγ complex. Both the Gα-GTP and the Gβγ dimer can interact with and regulate downstream targets, such as enzymes and ion channels, leading to varied cellular outcomes. The GTPγS binding assay is employed to measure the efficacy of synthetic cannabinoids in activating CBRs. It assesses the activation of G proteins by detecting the binding of GTPγS, a non-hydrolyzable form of GTP, to the G protein, thus allowing for the evaluation of synthetic cannabinoids’ potency.

The cAMP signaling pathway is another crucial mechanism by which CBRs exert their biological effects, being a natural energy sensor in mammalian cells that plays a key role in cellular and systemic energy homeostasis [45,46]. CBRs influence cAMP levels by modulating the activity of adenylyl cyclase, the enzyme that converts adenosine triphosphate (ATP) into cAMP. Depending on the subtype of the receptor and the type of cell it is expressed in, CBR activation can result in either an inhibition or a stimulation of adenylyl cyclase. An inhibition of this enzyme leads to a decrease in cAMP levels, while stimulation results in increased cAMP levels. These alterations in cAMP concentration affect protein kinase A (PKA) activity and other downstream targets, thereby influencing various cellular processes. The cAMP assay is a technique used to detect changes in intracellular cAMP concentrations following exposure to synthetic cannabinoids. It enables the investigation of cannabinoids’ effects on adenylyl cyclase activity and cAMP production, helping to determine whether these compounds act as agonists or antagonists at CBRs.

Beyond G protein-dependent mechanisms, CBRs can also signal through β-arrestin [47], which are multifunctional scaffolding proteins. Upon activation by cannabinoids, CBRs recruit β-arrestins, triggering receptor phosphorylation and internalization through clathrin-coated pits. β-arrestins are not merely involved in receptor desensitization and internalization but can also facilitate G protein-independent signaling pathways, leading to the activation of cellular responses such as mitogen-activated protein kinase (MAPK) pathways and gene transcription. A thorough understanding of these signaling routes—GTP, cAMP, and β-arrestin pathways—is fundamental for comprehending the physiological impact of cannabinoids and advancing the development of new therapeutic agents that target CBRs. Assays that focus on these pathways are integral to cannabinoid research, providing insight into the pharmacodynamics of synthetic cannabinoids and their interaction with CBRs. One such technique is the β-arrestin Recruitment Assay, which measures the engagement of β-arrestins with activated CBRs. This process, induced by synthetic cannabinoids, facilitates receptor internalization and initiates downstream signaling. The assay is instrumental in characterizing the biased signaling profile of cannabinoids, which may inform their therapeutic potential.

### 2.3. Cannabinoids

Cannabinoids from the cannabis plant, including Δ9-THC, CBD, cannbigerol (CBG), and D9-tetrahydrocannabivarin (THCV), exert effects by engaging with the ECS, specifically through CB_1_R in the central nervous system and CB_2_R in peripheral tissues and immune cells [48]. These interactions modulate signaling pathways, exhibiting anti-inflammatory properties by dampening immune responses and mitigating neuroinflammation—a promising avenue for treating chronic inflammatory conditions like neurodegenerative diseases. Neuroprotection is another notable cannabinoid effect, with several factors, including Δ9-THC counteracting the breakdown of tryptophan and improving conditions related to inflammation as well as showing strong antioxidant and anti-inflammatory properties advantageous in alleviating neuronal inflammation [49,50]; CBD exhibiting anti-inflammatory effects in a mouse model of AD induced by injection of human Aβ into the hippocampus as well as properties of reducing Aβ-induced GFAP (glial fibrillary acidic protein) mRNA, inducible NOS (iNOS) and interleukin IL-1β protein expression [51] and protecting against amphetamine-induced oxidative damage and increased brain-derived neurotrophic factor (BDNF) expression in a rat model of mania [52]; CBG predominantly exerting anti-inflammatory effects in vivo or have the ability to stimulate the degradation and removal of preformed, aggregated Aβ from neurons [53,54,55]; and THCV [56] indirectly and potentially (since THCV is an inverse agonist/selective antagonist of the CB1 receptor, it is similar to rimonabant but it does not have the identified adverse effects of rimonabant, and, in parallel, obesity-associated alterations in gut microbiota composition and function contribute to neuroinflammation, implicating a microbiological link between metabolic dysfunction and neurological consequences) implicated in safeguarding neuronal function (neuroprotection). Particularly, CBG demonstrates anti-inflammatory potential in preclinical models, distinct from CBD’s antagonistic affinity for GPR55, a GPCR involved in cellular processes like proliferation and differentiation. The antioxidant capabilities of cannabinoids, especially CBD, are vital for neutralizing free radicals and reducing oxidative stress, which is associated with neurodegenerative diseases. They also modulate neurotransmitter release, influencing a range of functions from pain relief to antiemetic and antispasmodic actions. Moreover, since the modulation of calcium distribution as well as mitochondrial function has an impact on metabolic impairments (linked frequently with neurodegenerative disorder progression) and, as Angelats et al. (2018) [57] indicated, CB_1_R mediates the cross-talk between calcium and cAMP signaling, it can be assumed that CBD’s role in maintaining calcium homeostasis is significant in preventing protein misfolding [58,59], which could be beneficial in conditions like Dravet syndrome [60] and PD [61]. Recent findings suggest Δ9-THC’s utility in reducing complications post-organ transplantation and in graft-versus-host disease (GVHD), influencing transplant policies in regions like California. Furthermore, the endogenous production of AEA and other endocannabinoids, linked to dietary omega fatty acids, indicates that dietary factors might affect cannabinoid receptor signaling. These multifaceted interactions of cannabinoids with the ECS and other cellular systems underline the importance of understanding their complex pharmacology to harness their full therapeutic potential and manage side effects.

#### Allosteric Modulation in Modulating Cannabinoid Receptor Function

Recent advancements in cannabinoid research have highlighted the importance of allosteric modulation as a nuanced means of influencing cannabinoid receptor function. Allosteric modulators are distinct from traditional orthosteric ligands as they bind to a different site on the receptor—known as the allosteric site—thereby modifying the receptor’s response to the primary (orthosteric) ligand. This modulation does not directly trigger receptor activation or inhibition but rather fine-tunes the receptor’s signaling capacity. There are several types of allosteric modulators: Positive allosteric modulators (PAMs) enhance the effects of orthosteric ligands, increasing receptor signaling. Negative allosteric modulators (NAMs) diminish these effects, leading to a decrease in receptor activity. Additionally, there are allosteric agonists, which can activate the receptor independently, and allosteric antagonists, which obstruct receptor activation by targeting the allosteric site [62,63]. While the research into allosteric modulators for CBRs is still emerging, it holds considerable promise. The development of CBR-specific allosteric modulators has been slower than for other GPCRs, but preclinical studies have begun to identify potential compounds that may offer targeted therapeutic benefits.

Org27569 [64] has been characterized as a PAM of CB_1_R, effectively potentiating the effects of the endogenous cannabinoid AEA both in vitro and in vivo. This modulator enhances the binding and efficacy of AEA without directly activating the receptor. In October 2019, significant insights into the structural interactions of CB_1_R were revealed with the publication of the first X-ray crystal structure of a ternary CB_1_R complex (PDB ID: 6KQI). This structure includes the orthosteric agonist CP55940 alongside Org27569, previously identified as a NAM. A computational analysis initially identified a site known as SC3—located at the intracellular junction near transmembrane helices 2, 6, and 7, and helix 8—as the most energetically favorable binding domain for Org27569. Contrary to these computational predictions, the X-ray crystal structure indicated that Org27569 binds at the SC6 site. This site had been determined by prior computational studies—using computational docking, Glide docking scores, binding free energy calculations, and extensive molecular dynamics (MD) simulations—to be the second most likely binding site for Org27569. These findings highlight the dynamic nature of receptor–ligand interactions and the crucial role of structural studies in verifying theoretical models [65].

GAT211 has been characterized as a PAM of the cannabinoid receptor CB_1_R. This compound shows promise for potential therapeutic applications in treating anxiety and depression. By enhancing the effects of both endogenous and exogenous cannabinoids, GAT211 positively modulates CB_1_R signaling. This modulation could lead to improved outcomes in managing mood disorders, demonstrating the significance of targeting allosteric sites on cannabinoid receptors for novel psychiatric medications [66].

GAT229 is recognized as a positive allosteric modulator of the cannabinoid receptor CB_1_R, demonstrating significant potential for pain management applications. This modulator enhances the signaling of endogenous cannabinoids at CB_1_R, which has been shown to produce analgesic effects in preclinical pain models. Such findings underscore the potential of GAT229 to contribute to the development of new pain relief therapies by leveraging the body’s natural cannabinoid systems [67].

Allosteric modulation of CB_1_Rs could offer an alternative strategy for treating diseases involving these receptors, potentially avoiding the undesirable effects linked to orthosteric stimulation. PSNCBAM-1 is an allosteric modulator of CB_1_R, with a distinct mechanism of action as NAM. It functions by reducing the affinity and signaling efficacy of orthosteric ligands, such as Δ9-THC, at CB_1_Rs. This modulation is particularly valuable in medical conditions associated with excessive CB_1_R activation, providing a targeted approach to mitigating adverse effects of cannabinoid overactivity. The unique properties of PSNCBAM-1 highlight its potential therapeutic applications in managing disorders linked to cannabinoid receptor dysregulation [68,69].

Current research is focused on identifying NAMs that selectively inhibit CB_1_R signaling, offering therapeutic potential for conditions linked to excessive CB_1_R activity, such as obesity, metabolic syndrome, and substance abuse disorders. In parallel, Group I metabotropic glutamate receptors, specifically mGlu1 (regions like the cerebellum, hippocampus, and other parts of the brain involved in cognitive and motor control—including learning and memory; mGlu1 receptors have been studied for their potential roles in disorders such as anxiety, chronic pain, and neurodegenerative diseases like PD) and mGlu5 (widely distributed throughout the brain, including the hippocampus, cortex, and basal ganglia, areas involved in cognition, perception, and motor control; mGlu5 receptors play critical roles in synaptic plasticity, a fundamental mechanism for learning and memory), are being explored for their potential in treating psychiatric and neurodegenerative diseases. The dual-target NAM VU0467558 has been investigated for its ability to simultaneously bind to mGlu1 and mGlu5, confirming crucial residues that contribute to receptor selectivity and dual binding [70]. Additionally, research involving the specific mGluR1 NAM, JNJ16259685, has shown promising results in reducing BBB permeability and cerebral edema following experimental subarachnoid hemorrhage (SAH). Post-SAH, glutamate is known to decrease levels and the phosphorylation of vasodilator-stimulated phosphoprotein (VASP), reduce occludin (a tight junction protein), and increase aquaporin-4 expression. The administration of JNJ16259685, however, has been observed to significantly elevate VASP and phosphorylated VASP levels, enhance occludin expression, and decrease aquaporin-4 expression at 72 h post-SAH [71].

ZCZ011 is characterized as a positive allosteric modulator of CB_1_R, with promising results in preclinical studies for treating pain and addiction-related behaviors. It binds to a specific site on the receptor, located outside the helical core and formed by the transmembrane domains TM2, TM3, and TM4. Advanced molecular dynamics simulations and mutagenesis experiments have revealed that the allosteric effects of ZCZ011 are mediated through a crucial rearrangement of TM2. This rearrangement enhances receptor activation by promoting a higher proportion of receptors to adopt an active conformation. In contrast, NAM ORG27569 also targets the same TM2-TM3-TM4 surface on CB_1_R but inhibits the rearrangement of TM2, thereby suppressing receptor activation and manifesting its NAM properties [72].

The discovery of allosteric modulators for CBRs opens new possibilities in drug development by potentially allowing for more precise control over receptor signaling than traditional orthosteric ligands. Allosteric modulators can selectively influence specific signaling pathways downstream of CBRs, paving the way for the creation of novel therapeutics that might offer improved efficacy and fewer side effects. In the context of receptor aggregation, allosteric modulators targeting CBRs could significantly affect receptor clustering or oligomerization, which in turn impacts key cellular processes such as receptor trafficking, signaling, and internalization. For example, allosteric modulators that stabilize receptor dimers or higher-order oligomers could enhance signaling pathways, intensifying therapeutic effects. Conversely, modulators that disrupt receptor aggregation may reduce signaling intensity, which could be beneficial in conditions where diminished receptor activity is desired.

### 2.4. Potential of Cannabinoids in Mitigating Protein Aggregation and Microglial Function

In neurodegenerative disorders, two key aspects have taken center stage in cannabinoid research: protein aggregation and the modulation of microglial function.

Protein aggregation, characterized by misfolded proteins clumping into insoluble structures, is central to the pathology of neurodegenerative diseases such as AD and PD. Understanding the mechanisms underlying protein aggregation is crucial for developing interventions to mitigate these degenerative conditions [73,74]. CBD, a non-psychoactive component of cannabis, has attracted significant attention for its broad pharmacological properties, particularly its potential for neuroprotection. CBD’s anti-inflammatory effects are particularly relevant, as chronic inflammation is closely linked to the progression of protein aggregation. By modulating immune responses and reducing neuroinflammation, CBD could indirectly influence protein aggregation and microglial activation, common features across various neurodegenerative diseases such as HD, ALS, AD, PD, and MS. CBD’s mechanisms include the suppression of pro-inflammatory cytokines like IL-6, TNF-alpha, and IL-1β, and the promotion of anti-inflammatory cytokines like IL-10. This cannabinoid also affects neurotransmitter systems, including serotonin and glutamate, which are crucial in neuroinflammation and neurodegeneration. By regulating neurotransmitter release and signaling, CBD may control synaptic transmission and neuronal excitability, further contributing to neuroprotection. Moreover, CBD can modulate microglial activation, promoting a shift from a pro-inflammatory M1 state to an anti-inflammatory M2 state, thereby reducing the release of pro-inflammatory cytokines and enhancing the production of anti-inflammatory mediators like TGF-beta. Additionally, CBD has been shown to suppress NF-κB activation and reduce the expression of enzymes like iNOS and COX-2, which are involved in neuroinflammation. Its antioxidant properties further protect neuronal cells from oxidative stress by scavenging free radicals and boosting endogenous antioxidant defenses [75,76]. CBD also impacts calcium homeostasis, which is critical for preventing excitotoxicity and promoting neuronal survival in models of neurodegenerative diseases [57]. The broader cannabis plant offers additional cannabinoids that may have synergistic effects, enhancing the therapeutic potential through what is known as the ‘entourage effect’. This suggests that a combination of multiple cannabis compounds might be more effective in treating neurodegenerative processes, including protein aggregation, than individual components alone [77]. While the preclinical data are promising, rigorous clinical trials are essential to determine the safety and efficacy of cannabinoids, including CBD, in treating or preventing neurodegenerative diseases. As research progresses, a deeper understanding of how cannabinoids influence protein aggregation and other neurodegenerative processes will emerge, potentially leading to novel therapeutic strategies.

Microglia, the resident immune cells of the CNS, play a critical role in maintaining neural homeostasis and responding to neuroinflammation. The dysregulation of microglial activity is a key factor in the pathogenesis of various neurodegenerative disorders such as HD, ALS, AD, PD, and MS. Importantly, the activation of CB_2_R has been shown to induce a shift in the microglial phenotype towards an anti-inflammatory state, thereby reducing the release of pro-inflammatory cytokines and neurotoxic factors, which underscores the therapeutic potential of cannabinoids [78,79,80]. When activated, microglia undergo morphological changes and release pro-inflammatory cytokines, chemokines, and reactive oxygen species (ROS). They also engage in phagocytosis to clear protein aggregates and damaged cells. Cannabinoids, by modulating these microglial responses, offer a multifaceted approach to treating neurodegenerative diseases [81,82]. This strategy not only alleviates neuroinflammation but also addresses the underlying molecular pathology that drives neuronal damage. Recent studies have highlighted the dynamic interaction between cannabinoids and microglial function, particularly through both CB_1_R and CB_2_R. The activation of these receptors triggers intracellular signaling pathways that regulate microglial activity, including cytokine release, phagocytosis, and migration, leading to neuroprotective outcomes [83,84]. Furthermore, cannabinoids are known to promote neurogenesis and neuroplasticity, processes essential for neuronal repair and regeneration. This not only helps counteract neuronal loss and protein aggregation but also improves synaptic function and enhances neuroregeneration. Cannabinoids reduce spasticity in animal studies, though the mechanisms are not fully understood. They inhibit presynaptic glutamate release and activate CB_1_R, decreasing glutamatergic transmission in animals treated with Δ9-THC. Chronic cannabis use also lowers glutamate metabolites in human brains. Dysregulated endocannabinoids are found in spasticity models and cerebrospinal fluid of patients with multiple sclerosis. Cannabinoid antagonists worsen spasticity, indicating a role for endocannabinoids in muscle tone regulation [81,82,85,86,87,88]. See Appendix A: Table A1.

### 2.5. Potential of Cannabinoids in Mitigating Protein Aggregation in HD, ALS, AD, PD, and SM

HD is a progressive neurodegenerative disorder marked by the accumulation of mutant huntingtin protein (mHTT), which leads to progressive motor dysfunction and cognitive decline. The pathological hallmark of HD is the formation of insoluble mHTT aggregates, notably intracellular inclusion bodies, which disrupt cellular processes and contribute to disease progression [89,90]. The aggregation of mHTT is driven by several factors, including impaired protein clearance, abnormal protein folding, and defective autophagy. Understanding these mechanisms is essential for developing effective therapeutic strategies. Cannabinoids have emerged as potential modulators of these pathways, showing promise in preclinical studies. By activating cannabinoid receptors CB_1_R and CB_2_R, cannabinoids can mitigate neuroinflammation, a significant contributor to HD progression and mHTT aggregation. Moreover, cannabinoids enhance the functionality of protein degradation systems such as the ubiquitin–proteasome system (UPS) and autophagy. The UPS degrades short-lived and misfolded proteins, while autophagy helps clear protein aggregates and damaged organelles through lysosomal degradation. Cannabinoids upregulate the expression and activity of components involved in these systems, promoting the clearance of mHTT aggregates in neuronal cells. Animal models of HD (studies in R6/2 mice and 3-nitropropionate-lesioned mice) treated with cannabinoids have shown reduced neuroinflammation, oxidative stress, and mHTT aggregation, alongside improvements in motor functions [91,92]—CBG was extremely active as a neuroprotectant in mice intoxicated with 3-nitropropionate: motor improvement by preserving striatal neurons against 3-nitropropionate toxicity, the attenuation of the reactive microgliosis and the upregulation of pro-inflammatory markers induced by 3-nitropropionate, and improvement in the levels of antioxidant defenses that were also significantly reduced by 3-nitropropionate [93]. These findings highlight the potential of cannabinoids to act as a co-therapeutic strategy in managing HD by targeting multiple pathological pathways, including inflammation, oxidative stress, and protein aggregation, with the potential to be used with a small molecular application—patients with HD show a notable decrease in CB1 receptor levels in MSNs of the caudate and putamen at early stages of the disease [94], Pérez-Arancibia et al. (2023) in study [95] noticed that common molecular pathways seem to be interesting targets to evaluate and develop new small molecules that modulate specific molecular processes including protein aggregation, inflammation, and G protein-coupled receptor signaling. Nevertheless, current clinical trials in neurological motor disorders have a high failure rate [95]. The reason might be hidden inside complexity and diversity, both genetic backgrounds and exposure of patients to environmental factors, which might influence the beginning of symptoms’ manifestation—small molecules have emerged as potential therapeutics for treating HD: advantages of using small molecules are small size, longer shelf life, cost-effective production, ease of modification, oral bioavailability, high pharmacological properties, and capability of crossing BBB [96]. While the preclinical data are promising, transitioning to clinical trials is crucial to validate the efficacy and safety of cannabinoids in the treatment of HD. This step will determine whether the observed benefits in animal models translate to human patients, potentially offering a multifaceted approach to combat the complex mechanisms underlying mHTT aggregation.

ALS is a progressive neurodegenerative disorder characterized by the deterioration of motor neurons, leading to muscle wasting and paralysis. While approximately 20% of ALS cases are hereditary, often linked to mutations such as those found in the superoxide dismutase-1 (SOD-1) gene, another significant pathological hallmark is the involvement of transactive response DNA-binding protein 43 (TDP-43). Recent research has highlighted the neuroprotective potential of CB_2_R activation in this context. A 2021 study utilizing TDP-43 transgenic mice investigated how CB_2_R activation affects ALS progression [97]. The results demonstrated that activating CB_2_R led to significant neuroprotective effects. Mice with an intact CB_2_R gene displayed improved motor performance on rotarod tests and experienced a slower neurological decline compared to their double-mutant counterparts. Morphological analyses of spinal cords revealed that motor neurons in double-mutant mice underwent premature demise, whereas those in CB_2_R-expressing transgenic mice maintained normal morphological characteristics. The study also assessed glial reactivity through GFAP and Iba-1 immunostaining, noting early glial reactivity in double-mutant mice at 65 days, which was absent in CB_2_R-expressing transgenic mice. By 90 days, both genotypes showed glial changes; however, CB_2_R-expressing mice exhibited more pronounced abnormalities in surviving motor neurons. The accelerated decline observed in double mutants resulted in premature mortality, whereas those expressing the CB_2_R had a more prolonged survival. Furthermore, the pharmacological deactivation of CB_2_R using the selective antagonist AM630 indicated a subtle exacerbation of decline in CB_2_R-expressing TDP-43 transgenic mice, underscoring the potential of CB_2_R agonists as a neuroprotective strategy in ALS management.

Cx43, a key protein involved in gap junction communication between neighboring cells, is increasingly recognized for its role in the pathophysiology of neurodegenerative diseases like AD, PD, and ischemic brain injury. An altered expression and functionality of Cx43 have been implicated in these disorders, presenting a novel target for therapeutic intervention. Sáez et al. (2018) [98] as well as Gajardo-Gómez et al. (2017) [99] suggest that cannabinoids, interacting with the endocannabinoid system encompassing CB_1_R and CB_2_R, may help normalize Cx43 activity. This interaction could be particularly beneficial in neurodegenerative diseases where disrupted cellular communication contributes to disease progression. Cannabinoids are known for their anti-inflammatory and neuroprotective properties. In conditions characterized by chronic inflammation, such as AD, cannabinoids could reduce inflammation, indirectly facilitating the regulation of Cx43 and improving intercellular communication. This normalization of Cx43 function by cannabinoids may enhance neuronal connectivity and neuroprotection, potentially slowing the progression of neurodegenerative diseases. The therapeutic implications of modulating Cx43 with cannabinoids could be profound, improving cognitive functions linked to hippocampal activity and reducing neurotoxicity associated with abnormal protein aggregates, such as Aβ plaques. Specifically, cannabinoids like Δ9-THC have shown promise in inhibiting Aβ plaque formation by modulating enzymes involved in Aβ production and clearance. In experimental models, cannabinoids have reduced the activation of astrocytic Cx43 hemichannels, decreasing the release of harmful gliotransmitters and protecting against Aβ-induced damage in hippocampal cells. Research, carried out by Ren et al. (2018) [100], involving genetically modified animal models, such as an astroglial Cx43 knockout (KO) AD mouse model being the result of crossbreeding GFAP-Cx43 KO mice with APP/PS1 mice, has demonstrated preserved cognitive functions, underscoring the potential benefits of targeting Cx43 in AD therapy (also review [101] indicates that treatment with cannabinoids has shown the ability of reducing the activation of astrocytic Cx43 hemichannels as well as of reducing the release of gliotransmitters (excitotoxic glutamate and ATP) from astrocytes). Moreover, cannabinoids reduced the release of gliotransmitters (excitotoxic glutamate and ATP) from astrocytes and prevented β-amyloid-induced neuronal damage in the hippocampal slices. Furthermore, Δ9-THC’s interactions with Aβ peptides suggest a dose-dependent anti-aggregation effect, which might be leveraged to mitigate AD pathology more effectively than current AChE inhibitors. Research [102] by Vik-Mo et al. (2020) suggests a potential role of TDP-43 in aggression in AD, indicating its association with neuropsychiatric symptoms. Cannabinoids, particularly CBD, have been investigated for their effects on TDP-43 pathways, offering a potential avenue for managing symptoms associated with TDP-43 pathology in dementia. Studies [101] by He et al. (2020) and [103] by Hampel et al. (2021) emphasize the excessive accumulation of proteins in AD and other dementias, exacerbating inflammatory processes [104]—it is highlighted that receptors on microglia can bind Aβ fibrils, driving an inflammatory response similar to the M1 (pro-inflammatory) phenotype observed outside the central nervous system [105]; it was reported that cellular prion protein (PrPC) is one of the most selective and high-affinity binding partners of Aβ oligomers. The interaction of Aβ oligomers with PrPC is important to synaptic dysfunction and loss. The results demonstrate that PrPC accumulates with aging in human brain tissue even prior to AD mainly within diffuse-type amyloid plaques.

PD is marked by the aggregation of alpha-synuclein (α-synuclein) within Lewy bodies, leading to progressive neurodegeneration. An emerging investigation is exploring how cannabinoids might influence this process, with particular interest in their interaction with the endocannabinoid system and α-synuclein. Recent studies have shown complex dynamics between cannabinoids and α-synuclein. Research carried out by Kelly et al. (2022) [106] involving α-synuclein overexpression in rat models identified a dysregulation of the endocannabinoid system, specifically noting a decrease in the endocannabinoid 2-AG and the lipid mediator oleoylethanolamide (OEA). Despite this reduction, neuroinflammatory markers were concurrently diminished, suggesting a nuanced bidirectional regulation between α-synuclein and endocannabinoid levels. Further investigations into PD models have revealed neuroprotective effects of Δ9-THC and CBD. In a 6-OHDA model of PD, both cannabinoids were shown to reduce the loss of dopaminergic neurons [107]. Remarkably, CBD increased dopamine concentration in the striatum independently of cannabinoid receptors CB_1_R and CB_2_R, indicating potential receptor-independent neuroprotective pathways. Moreover, a 2023 study by da Cruz Guedes et al. [108] utilizing Caenorhabditis elegans models of PD found that CBD could reverse locomotor deficits induced by reserpine and α-synuclein. The study noted that CBD’s effects on behavior, such as reductions in pharyngeal pumping, did not alter other physiological functions like defecation or egg production, hinting at specific therapeutic targets. The research suggested that endocannabinoids, particularly 2-AG and AEA, and interactions with the NPR-19 receptor might play roles in neuroprotection, possibly influenced by dietary factors. CBD’s broad receptor activity, including low-affinity interactions with CB_1_R and CB_2_R, as well as engagement with serotonin 5-HT1A and vanilloid receptors TRPV1 and TRPV2, underscores its pleiotropic nature. The proposed mechanisms involve the modulation of dopamine release, activation of TRPV1- and TRPN-like channels, and protection against oxidative stress and α-synuclein accumulation in dopaminergic neurons. These findings illustrate a multifaceted landscape where cannabinoids, especially CBD, may offer neuroprotective advantages in PD by modulating α-synuclein pathology. While current results are promising, further research is essential to fully understand these interactions and develop targeted cannabinoid-based therapeutic strategies for PD.

MS is a chronic autoimmune disorder of the CNS, marked by inflammation, demyelination, and neurodegeneration [93]. In MS, immunological processes lead to the aggregation of autoreactive immune cells, such as T lymphocytes, which target self-antigens like myelin proteins. This results in perivascular inflammatory infiltrates and the formation of demyelinating lesions, central to the disease’s pathogenesis. Neuronal and axonal damage, another hallmark of MS, involves the accumulation of compromised neural elements in areas of active inflammation, contributing to the progressive disability seen in patients with MS. Additionally, emerging research points to the presence of protein aggregates in the CNS, which may further complicate MS pathology by intertwining neuroinflammatory and neurodegenerative processes [93]. Cannabinoids, active compounds from the cannabis plant, are being investigated for their therapeutic potential in MS. These compounds interact with the endocannabinoid system, which plays a crucial role in maintaining homeostasis, modulating immune responses, and protecting neuronal integrity. Studies have shown that cannabinoids possess immunomodulatory and anti-inflammatory properties [93,95], potentially mitigating the autoimmune and inflammatory components of MS. Through the modulation of immune cell activity, cannabinoids influence the immunopathological landscape of MS, and their antioxidative and anti-excitotoxic properties contribute to neuroprotection. Moreover, cannabinoids may promote neurorepair and remyelination, addressing both inflammatory and neurodegenerative aspects of MS. Their role in modulating protein homeostasis is particularly compelling, as they may influence the formation and clearance of protein aggregates within the CNS. Integrating cannabinoid-based therapies in MS treatment could offer a novel approach to manage this complex disorder. This would not only aim to improve clinical outcomes but also enhance the quality of life for those affected. As research progresses, understanding the precise mechanisms by which cannabinoids affect protein aggregation, microglial responses, and overall neurodegeneration will be crucial.

Vitamin B12, or cobalamin, which is essential for DNA synthesis, red blood cell formation, neurological function, and the metabolism of fatty acids and amino acids, might also interact beneficially with cannabinoids. This potential synergy is intriguing, given the positive effects observed with omega-3 fatty acids in managing conditions like ALS [2], and the enhanced efficacy of CBD when combined with selenium in reducing cancer tissues [109]. Furthermore, the role of microglia in maintaining CNS homeostasis and their response to neurodegenerative stressors highlights the importance of exploring combinations of cannabinoids with vitamin B12. Microglia respond to stress signals, such as protein aggregates, by altering their phenotype, which includes changes in morphology and an increase in immune-related molecules. Vitamin B12 is known for its neuroprotective properties, particularly in reducing inflammation and promoting the clearance of protein aggregates. These attributes, combined with the regulatory effects of cannabinoids on neuroinflammation and microglial activation, suggest that a theoretical combination of vitamin B12 and cannabinoids could offer enhanced therapeutic benefits in neurodegenerative diseases. The upcoming section will explore this impact of vitamin B12. See Appendix A: Table A2.

## 3. Impact of Vitamin B12 in Neurodegenerative Processes. Therapeutic Potential of Vitamin B12 Combination with Cannabinoids in Neurodegeneration

Advancing the holistic understanding of neurodegenerative interventions requires a nuanced exploration of synergies between cannabinoids and vitamin B12. Building upon the foundation laid by the investigation of cannabinoids in microglial impairment, this scientific inquiry takes a pivotal turn towards a broader spectrum of interventions, with a specific focus on the multifaceted impact of cannabinoids on microglial function [11]. The inclusion of vitamin B12 as an adjunctive discourse further enriches our understanding of comprehensive neurodegenerative strategies. Vitamin B12, also known as cobalamin, plays a crucial role in various physiological processes within the human body, including DNA synthesis, red blood cell formation, and maintenance of the nervous system including the production of myelin, the insulation around nerve fibers that facilitates proper nerve signal transmission. Deficiencies in vitamin B12 have been associated with neurological disorders, including peripheral neuropathy and cognitive decline. Its involvement in neurodegenerative disorders has been a subject of scientific investigation, particularly in disorders such as AD and PD. While the direct causative relationship between vitamin B12 deficiency and these neurodegenerative disorders is not fully elucidated, there is evidence to suggest that inadequate levels of vitamin B12 may contribute to their development or exacerbate their symptoms. Vitamin B12 is essential for the conversion of homocysteine to methionine, a process facilitated by the enzyme methionine synthase. In the absence of adequate vitamin B12, this conversion is impaired, leading to elevated levels of homocysteine in the blood. Elevated homocysteine has been associated with increased risk of neurodegenerative disorders such as AD and vascular dementia. High homocysteine levels are believed to contribute to neuronal damage through oxidative stress, inflammation, and endothelial dysfunction, ultimately leading to neurodegeneration.

Neuroprotective Effects: Vitamin B12 exhibits neuroprotective effects through its involvement in various biochemical pathways, including the synthesis of neurotransmitters such as serotonin and dopamine. Adequate levels of vitamin B12 are necessary for optimal neuronal function and protection against oxidative stress and neurotoxicity. Deficiency in vitamin B12 may compromise these neuroprotective mechanisms, rendering neurons more susceptible to damage and contributing to the pathogenesis of neurodegenerative disorders. Vitamin B12 deficiency has been associated with microvascular changes and cerebral hypoperfusion, which can result in neuronal injury and contribute to the development of neurodegenerative disorders such as vascular dementia. Impaired cerebral blood flow due to microvascular changes may lead to hypoxic conditions in the brain, exacerbating neuronal damage and cognitive decline. While the evidence suggests a link between vitamin B12 deficiency and neurodegenerative disorders, it is essential to recognize that these disorders are multifactorial in nature, with genetic, environmental, and lifestyle factors also playing significant roles in their pathogenesis. Nonetheless, maintaining adequate levels of vitamin B12 through dietary intake or supplementation may have potential benefits in reducing the risk of neurodegenerative disorders and preserving neurological function. Further research is needed to elucidate the precise mechanisms underlying the connection between vitamin B12 and neurodegenerative diseases and to explore the potential therapeutic implications of vitamin B12 supplementation in these conditions. Vitamin B12 is highlighted as a cofactor in crucial cellular processes implicated in neurodegeneration. The correlation between the impact of cannabinoids on microglial impairment and the inclusion of vitamin B12 underscores the necessity of addressing neurodegenerative processes comprehensively. Studies emphasize the multifaceted role of vitamin B12 (as well as omega-3 fatty acids) in mental wellbeing, the reduction in oxidative stress markers, the regulation of lipid metabolism, and the potential to reduce the risk of cognitive decline [110,111,112,113].

Recent scientific investigations independently highlight the pivotal influence of vitamin B12 on various neurodegenerative processes. Its role in DNA synthesis, maintenance of neuronal myelin, cognitive health, vascular-mediated Aβ clearance, BBB integrity, protein aggregation pathways, and the modulation of microglial function is accentuated [9]. This wealth of insights contributes to a profound understanding of potential synergies between cannabinoids and vitamin B12 and highlight integrated therapeutic strategies. The insights from cannabinoids and vitamin B12 indicate a more comprehensive approach to neurodegenerative interventions. By combining insights from the impact of cannabinoids on microglial function with the multifaceted roles of vitamin B12, this integrative approach holds promise for developing comprehensive and synergistic strategies in the management of neurodegenerative disorders. Further research and clinical validation are essential to unlock the full therapeutic potential of this integrated paradigm. Deficiencies in vitamin B12 have been linked to neurodegenerative disorders, with supplementation exhibiting potential in mitigating certain facets of cognitive decline [9]. Emerging evidence suggests that vitamin B12 may extend its influence beyond traditional roles, encompassing novel pathways that contribute to the neurodegenerative challenges. Recently, study [10] proposed the superiority of B12-HD in fostering vascular-mediated Aβ clearance across the BBB. This aligns with the findings of [113], emphasizing the BBB as the primary route for Aβ clearance. The nuanced interplay between vitamin B12 and BBB integrity is crucial, shedding light on potential avenues for therapeutic interventions in neurodegenerative diseases. The aging process and neurodegenerative diseases induce modifications to the BBB. Notably, the loosening of tight junctions among endothelial cells contributes to increased capillary permeability and the influx of neurotoxic molecules. Alterations in the composition of collagens and laminins, coupled with microvascular fibrosis in the BBB basement membrane, result in heightened vascular basement membrane thickness and stiffness [8]. Understanding these modifications provides insights into the dynamic relationship between vitamin B12 and BBB integrity [114]. Emerging evidence suggests that vitamin B12 may play a role in influencing protein aggregation pathways. Its regulation of homocysteine levels, a precursor of S-adenosylmethionine (SAM) involved in protein methylation, underscores its potential impact on protein misfolding and aggregation observed in neurodegenerative diseases [115]. Microglial activation, influenced by oxidative stress and inflammation, could be attenuated by the antioxidant properties of vitamin B12 [11]. This dual influence on both protein aggregation and microglial function emphasizes the comprehensive nature of vitamin B12 in addressing the multifactorial aspects of neurodegeneration. This comprehensive review highlights the multifaceted impact of vitamin B12 in neurodegenerative processes. From its traditional roles in cognitive health to its involvement in vascular-mediated Aβ clearance, BBB integrity, protein aggregation pathways, and microglial modulation, vitamin B12 may be considered as a relevant therapeutic factor in the case of neurodegenerative disorders. Further research and clinical investigations are imperative to validate the therapeutic potential of vitamin B12 and its nuanced interactions. Moreover, elevated homocysteine levels, resulting from vitamin B12 deficiency, have been associated with an increased risk of AD. High homocysteine levels can contribute to vascular dysfunction, oxidative stress, and neuronal damage, all of which are implicated in the pathogenesis of AD. Vitamin B12 deficiency can lead to impaired methylation reactions, affecting the synthesis and maintenance of myelin sheaths. This can result in neuronal demyelination, disrupted nerve signal transmission, and cognitive impairment characteristic of AD. Adequate levels of vitamin B12 are necessary for optimal neuronal function and protection against neurotoxicity. Vitamin B12 deficiency may compromise these neuroprotective mechanisms, rendering neurons more vulnerable to Aβ deposition and tau protein pathology, key pathological features of AD. Vitamin B12 deficiency has been associated with microvascular changes and cerebral hypoperfusion, contributing to neuronal injury and cognitive decline observed in AD. Impaired cerebral blood flow may exacerbate the accumulation of toxic metabolites in the brain, further promoting neurodegeneration [116,117]. Elevated homocysteine levels resulting from vitamin B12 deficiency have been linked to an increased risk of PD. High homocysteine levels can induce oxidative stress, mitochondrial dysfunction, and dopaminergic neuron degeneration, contributing to the pathogenesis of PD. Vitamin B12 deficiency can impair methylation reactions within the nervous system, affecting the synthesis of neurotransmitters such as dopamine. Dopaminergic neuron dysfunction is a hallmark feature of PD, and vitamin B12 deficiency may exacerbate dopaminergic neuronal loss through impaired methylation processes. Adequate levels of vitamin B12 may help protect dopaminergic neurons from oxidative stress and neurotoxicity, potentially reducing the risk of PD development or progression. Vitamin B12 deficiency has been associated with microvascular changes and impaired cerebral blood flow, which may contribute to neuroinflammation and neuronal injury observed in PD. Impaired cerebral perfusion may exacerbate dopaminergic neuron degeneration and motor symptoms associated with PD [118,119,120].

Multiple Sclerosis: Vitamin B12 is essential for methylation reactions, including the methylation of myelin basic protein (MBP), a key component of myelin sheaths that insulate nerve fibers. Myelin sheaths facilitate the efficient transmission of nerve impulses. Disruption in methylation reactions due to vitamin B12 deficiency could impair myelin maintenance and repair processes, potentially exacerbating demyelination observed in MS lesions. Vitamin B12 has been shown to have immunomodulatory properties, including the regulation of immune cell function and cytokine production. The dysregulation of the immune system plays a central role in the pathogenesis of MS, with immune cells mistakenly attacking myelin and causing inflammation and neuronal damage. While the specific impact of vitamin B12 on MS-related immune dysregulation is not fully understood, it is possible that vitamin B12 deficiency could contribute to immune dysfunction and exacerbate the inflammatory processes involved in MS. In MS, oxidative stress contributes to neuronal damage and disease progression. Adequate levels of vitamin B12 may help mitigate oxidative stress and protect neurons from damage, potentially slowing the progression of MS-related neurodegeneration. Some patients with MS may experience symptoms such as fatigue, cognitive impairment, and depression, which can be associated with vitamin B12 deficiency. These symptoms can overlap with those of MS, making it challenging to distinguish between vitamin B12 deficiency-related symptoms and MS-related symptoms. Addressing vitamin B12 deficiency through supplementation may improve overall wellbeing and quality of life in patients with MS, although it may not directly impact the course of the disease [121,122].

Therapeutic Potential in Integrating CBD and Vitamin B12: Integrating various perspectives, including insights from [123] and additional information on the therapeutic effects of combining CBD with other compounds, presents a holistic framework for addressing neurodegeneration. Recent research, carried out by Jonnalagadda et al. (2023), as highlighted in [123], emphasizes the influence of vitamin B12 on neurodegenerative processes. Fingolimod, a medication used in the treatment of multiple sclerosis, is found to facilitate the maintenance of vitamin B12 homeostasis in the CNS. Fingolimod achieves this by directly binding to Transcobalamin 2 (TCN_2_), forming a complex that enhances the availability of vitamin B12 in astrocytes. This process involves the internalization of CD320, followed by the phosphorylation of fingolimod, which functionally antagonizes Sphingosine-1-Phosphate receptor 1 (S1P1), resulting in the upregulation of CD320 in astrocytes. Expanding on this narrative, studies exploring the inclusion of CBD as an additive compound in therapy have demonstrated an increased therapeutic effect when combined with specific compounds; for instance, in the combination with selenium, the CBD + Nano-Se group exhibited a significantly higher number of capillaries in chicken muscles compared to the control group or selenium alone (*p* < 0.05). This combination also resulted in a reduced number of necrotic muscle fibers in chickens infected with *C. perfringens* [109]. This enhanced effect of including CBD is crucial not only from a cancer perspective but also in the context of neurodegenerative diseases, as corroborated by studies on selenium in HD treatment. Selenium microparticles were found to significantly reduce neuronal degradation, decrease behavioral dysfunctions, and protect Caenorhabditis elegans against stress-induced damage. The molecular mechanism revealed that Nano-Se attenuated oxidative stress, inhibited huntingtin protein aggregation, and downregulated the expression of histone deacetylase family members at mRNA levels [124]. Additionally, research on the sigma-1 receptor as a protective factor for diabetes-related cognitive dysfunction, involving the regulation of astrocyte endoplasmic reticulum–mitochondria contact and endoplasmic reticulum stress, is worth mentioning [109]. There is also an observation regarding the possible susceptibility in elderly patients with HD to the development of ALS features with an atypical distribution of TDP-43 that resembles aggregated mutant huntingtin. This comprehensive exploration of vitamin B12, CBD, and other compounds enriches understanding of interconnected pathways in neurodegenerative processes. As research progresses, the potential for integrated therapeutic strategies encompassing cannabinoids, vitamin B12, and other factors offers a promising avenue for advancing the treatment of neurodegenerative diseases.

### Dual-Targeted Vitamin B12 Liposomes and Cannabidiol in Micellar Systems

Wu et al. (2020) [125] explored the progression of ALS with a focus on the BBB. Their findings indicated that cerebrospinal fluid (CSF) homocysteine (Hcy) levels and the albumin quotient (Qalb) were substantially higher in patients with ALS compared to controls. The CSF Hcy levels were 0.50 ± 0.46 μmol/L in patients with ALS, versus 0.25 ± 0.27 μmol/L in the control group, while Qalb was 8.09 ± 3.03 in patients with ALS and 6.39 ± 2.21 in controls, with both differences showing statistical significance (*p* < 0.05). A further analysis using a generalized linear mixed model revealed a positive correlation between CSF Hcy and Qalb in patients with ALS (*p* < 0.05). Homocysteine, a redox-active amino acid, has been linked to various vascular diseases due to its pro-inflammatory nature and its ability to promote oxidative stress, contributing to neurotoxicity. High Hcy levels in CSF or plasma have also been associated with neurodegenerative diseases like PD and AD. The integrity of the BBB plays a crucial role in brain health, and its breakdown has been implicated in early motor neuron degeneration in rodent models of ALS, such as in SOD1G93A rats. In these models, BBB disruption occurred before motor neuron damage, suggesting that BBB impairment might trigger ALS. Additionally, disruptions in the BBB/BSCB with reduced pericyte counts and diminished tight junction proteins have been observed in the spinal cords of patients with ALS. These alterations are often accompanied by oxidative stress and inflammation. Pericytes, sharing a basement membrane with endothelial cells, are integral to controlling BBB permeability. High levels of Hcy can also reduce cell viability and lead to brain damage through mechanisms like oxidative stress, apoptosis, and disrupted methylation. In the study, the CSF Hcy level was notably elevated in patients with ALS, while the plasma Hcy level did not show a significant difference between patients with ALS and controls. A Spearman correlation analysis demonstrated a positive, yet non-significant, correlation between CSF Hcy and Qalb in patients with ALS, with markedly distinct patterns of correlation among CSF and plasma/serum indices when comparing patients with ALS and control groups. These findings suggest that altered Hcy metabolism in patients with ALS may be linked to BBB dysfunction.

El-Mezayen et al., 2022 [10], indicate that vitamin B12 plays a pivotal role in neurodegenerative diseases, notably AD, through its impact on BBB and cholinergic systems. A study highlighting the connection between vitamin B12 and the BBB in the context of AD examined the limitations of cholinergic function in the brain, which pose a significant challenge for targeting cholinergic pathways effectively. The search for a disease-modifying agent that can increase brain cholinergic receptors while addressing various AD hallmarks has been a crucial area of research. Vitamin B12 functions as an epigenetic modifier with a unique transport system in CNS via cubilin receptors. These receptors contain an agrin protein, which is known to aggregate cholinergic components, suggesting that vitamin B12 administration may induce cholinergic receptor aggregation. This aggregation has the potential to counteract cholinergic loss in AD. Additionally, vitamin B12 is involved in homocysteine (Hcy) metabolism. Elevated levels of Hcy have been linked to disruptions in BBB integrity, a common feature in neurodegenerative conditions like AD. B12’s role in metabolizing Hcy may help restore BBB integrity, thereby offering a dual benefit for AD management. A pharmacological model of cholinergic amnesia compared three different B12 doses to the standard of care (donepezil, or DON) regarding cholinergic system modulation and their impact on Hcy metabolic pathways. The study also examined AD-related cerebrovascular pathology through morphometric analyses and ultrastructure assessments with scanning and transmission electron microscopes. The highest tested dose of vitamin B12 (B12-HD) showed the most significant hippocampal cholinergic modulation, with a dose-dependent increase in cholinergic receptor activity. Altered Hcy metabolism, a consequence of cholinergic disruption, was variably reversed by different B12 doses. While both DON and B12-HD therapies exhibited equivalent efficacy in reducing β-amyloid synthesis, the B12-HD-treated group demonstrated a superior ability to restore BBB integrity, suggesting a higher capacity for β-amyloid clearance. This research underscores the potential of vitamin B12 as a therapeutic agent in AD, especially in its ability to support BBB integrity and cholinergic function, providing a promising approach to treating or slowing the progression of AD.

The Andrade et al. (2022) [126] study focuses on the challenge of vitamin B12’s high molecular weight and hydrophilicity, which limit its ability to cross the BBB, potentially compromising its clinical application in neurodegenerative diseases. To address this issue, the researchers explored a dual-targeting approach using vitamin B12-loaded liposomes functionalized with transferrin (Tf), aiming to target both the BBB and neuronal cells. The rationale behind using transferrin is based on the overexpression of transferrin receptors in these cells, which facilitates the passage of vitamin B12 across the BBB. The study’s findings indicate that these vitamin B12-loaded liposomes had a size smaller than 200 nm, low polydispersity, and neutral zeta potential, making them suitable for brain delivery. The liposomes also showed adequate encapsulation efficiency and sustained release of vitamin B12 over a 9-day period, with physical stability maintained for up to 2 months under storage conditions. These characteristics suggest a promising delivery system for vitamin B12 in therapeutic applications, particularly for neurodegenerative diseases. To enhance the therapeutic potential of vitamin B12, combining it with CBD could be beneficial, especially in a micellar system [5]. Micellar systems are known for their high encapsulation efficiency, providing a viable platform for developing novel cannabidiol dosage forms for neurodegenerative disease treatments. This encapsulation system reached an encapsulation efficiency of up to 84%, offering a promising means of delivering hydrophobic drugs like CBD effectively. Cannabinoids, particularly those targeting the CB_2_R, play a role in neuro-organic environmental modulation, which is critical in neurodegenerative diseases. A successful formulation incorporating nanoparticles of β-caryophyllene [4], a CB_2_R binder, marks the completion of an initial phase in the development and optimization of a therapeutic prototype. This suggests that cannabinoids like CBD could potentially complement the therapeutic use of vitamin B12 by addressing the high-molecular-weight barrier, enabling a more effective treatment for neurodegenerative conditions. Combining these approaches—vitamin B12-loaded liposomes with transferrin and CBD in a micellar system—could lead to improved treatment outcomes in neurodegenerative diseases, overcoming the limitations of BBB crossing and enhancing the overall therapeutic impact.

Moreover, the potential advantages of vitamin B12 supplementation are particularly advantageous for patients experiencing reduced mobility and strength, who seek to tailor their treatment around their personal schedules. This research is a foundational effort to address the existing gaps in academic studies by advocating for further randomized controlled trials. These trials should focus on examining and elucidating the impact of vitamin B12 injections on disease progression, muscle functionality, and life quality in a small yet varied group of patients with ALS. The study conducted by Zubair et al. (2023) [127] identified 4 main themes and 11 subthemes from the collected data, which included initial conditions, the process of administering the injection, patients’ personal experiences with vitamin B12, and the outcomes and anticipations. The participants all acknowledged certain benefits from vitamin B12 injections, notably enhanced energy levels, decreased fatigue, and better balance. Nonetheless, certain patients encountered challenges in tracking the precise effects of the treatment due to the progressive nature of ALS. Most participants reported noticeable benefits, particularly in terms of energy, fatigue reduction, and balance. However, one individual highlighted the difficulty in assessing the effect of their treatment regimen because of the progressive symptoms of the disease. All but this patient noted a significant decline in energy and a discernible difference on days when B12 was not administered. A key limitation of the study is its limited sample size. This issue is compounded by the difficulty of conducting interviews with patients with ALS, many of whom may have speech challenges. See Appendix A: Table A3.

## 4. Conclusions

Since neurodegenerative diseases like Alzheimer’s, Parkinson’s, multiple sclerosis, Huntington’s, and amyotrophic lateral sclerosis present significant healthcare and therapeutic challenges due to not only their complex etiology or pathophysiology but symptoms severity as well, it is important to keep the attention on improving constantly effective therapeutic methods devoted to neurodegenerative diseases treatment. Recent studies indicate cannabinoids, particularly from Cannabis sativa, to hold promise in addressing key pathological processes associated with these disorders. Cannabinoids, especially THC and CBD, demonstrate anti-aggregative effects, modulating the endocannabinoid system and interacting with cannabinoid receptors 1 and 2, offering potential in mitigating protein aggregation seen in disorders like multiple sclerosis. They also activate CBR1, protecting against mitochondrial dysfunction, crucial in diseases disrupting energy distribution, such as demyelination. Emerging evidence suggests that vitamin B12, essential for cellular processes, could complement therapeutic strategies, potentially enhancing the effects of CBD. Additionally, CBD shows promise in reversing locomotor changes in Parkinson’s disease independently of NPR-19 receptors, while also protecting dopaminergic neurons and reducing reactive oxygen species accumulation. Thus, the integration of nanoparticles of β-caryophyllene, a CB2R binder, as explored by Alberti et al. (2020) [4], represents potential advancement in developing therapies that improve drug BBB crossing and enhance overall treatment efficacy, moreover, accordingly, the process aimed at combining RNA aptamers with cannabinoids and vitamin B12 may offer precise targeted therapies, but rigorous testing is necessary before clinical use. This combined approach represents a promising frontier in neurodegenerative disease treatment, highlighting ongoing research into cannabinoids’ effects and applications across various disease contexts. Understanding their interaction with mitochondrial function and cellular communication holds potential for novel therapeutic strategies. Further investigation is needed to fully grasp cannabinoids’ effects and applications in diverse disease contexts.

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
