# Peer review of "Cannabinoids: Potential for Modulation and Enhancement When Combined with Vitamin B12 in Case of Neurodegenerative Disorders"

_pharmaceuticals, 2024, doi:10.3390/ph17060813_

Round 1

Reviewer 1 Report (Previous Reviewer 2)

Comments and Suggestions for Authors

The document readability would be enhanced by adopting a more organized format to present the information. It is suggested that the document delve deeper into the specific mechanisms that may drive the synergistic relationship between cannabinoids and vitamin B12. Furthermore, the document could benefit from a more critical analysis of the limitations and challenges associated with the use of cannabinoids and vitamin B12 in neurodegenerative disorders.

The author is encouraged to provide more detailed explanations of the molecular mechanisms through which cannabinoids and vitamin B12 exert their therapeutic effects. For the sake of clarity, the use of tables to summarize these interactions for each neurodegenerative disease would greatly improve the document's readability.

In its current state, the document offers promising perspectives on the use of cannabinoids and vitamin B12 for treating neurodegenerative diseases. Yet, there is potential for the document to make a more significant impact with better structure, clearer exposition, and a more rigorous examination of the relevant academic work. Addressing these recommendations would enhance the document's contribution to the field of neurodegenerative disease intervention research.

Comments on the Quality of English Language

The English quality of the manuscript is good, requiring minor editing

Author Response

Dear Reviewer,

at the very beginning I would like to thank you truly for your support as well as honest and comprehensive feedback. However, I am not sure if I managed with everything I was supposed to.

Q: The document readability would be enhanced by adopting a more organized format to present the information. It is suggested that the document delve deeper into the specific mechanisms that may drive the synergistic relationship between cannabinoids and vitamin B12. Furthermore, the document could benefit from a more critical analysis of the limitations and challenges associated with the use of cannabinoids and vitamin B12 in neurodegenerative disorders.

A: According to the limitations and challenges associated with the use of cannabinoids and vitamin B12 in neurodegenerative disorders, I added table regarding this aspect, but I am not sure if it is comprehensive sufficiently. I was concerned a bit about manuscript length (even if I shortened it by reducing references to 126 as well as making corrections including shortening the Conclusion part).

Q: The author is encouraged to provide more detailed explanations of the molecular mechanisms through which cannabinoids and vitamin B12 exert their therapeutic effects. For the sake of clarity, the use of tables to summarize these interactions for each neurodegenerative disease would greatly improve the document's readability.

A: According to this comment, I added 3 tables (I made corrections mostly by preparing tables concerning suggested information, however some changes regarding additional information are marked in text using “track changes”), but I am not sure if the molecular mechanisms are described sufficiently - I will make corrections if needed.

Reviewer 2 Report (Previous Reviewer 1)

Comments and Suggestions for Authors

Author reviews the therapeutic potential of cannabinoids and vit B12. The review is lengthy, not well focused ad not sub-sectioned. It is necessary to revise this review, adding subsections as formatted by „instructions to authors” and „mdpi template”. Please number subsections.

It is suggested to shorten the longer subsections , e.g. related to connexins, MS, Vit B12, and also, it is not necessary to add an extra „Discussion „ section. Still, Conclusion section is lengthy, please focus.

Author uses as much as 152 references, but many of them do not match the referred text, and still, there are some discrepancies (see specific comments). Thus, the references in the text need an extensive review.

Specific comments

Please review Title (which seems to be 2 sentences)

Please explain all abbreviations at first mention.

Please name „Researchers” when cite them at the beginning of sentences (as Author et al.) , e.g. in lines 55,57,126,130, etc.

„Cannabinoid receptor CB1 (CBR1)” should be corrected as „cannabinoid type 1 receptor (CB1R)” and initials corrected to CB1R or CB2R throughout the text.

line 207, reference related to TRPA1 is missing

line 233, ref related to animal models is missing

line 242, please rephrase „debilitating conditions”

line 278, 337, refs are necessary

line 341-344: refs 43-44 are missing! From this point mismatches of references are frequent! For example, in line 367, ref. 49 should be ref. 47. In line 392: THC is diccussed from ref 50, CBD from ref 53, CBG from ref 55, THCV from ref 57, please check! Also please put references in numeric orders (e.g. 55,59 instead of 59,55).

lines 405-406 seems to be refs 63,64

line  420 seems to be ref 71 (not 73), ref 72 (not 74)

line 445 seems to be refs 75-76, line 450 maybe ref 77, line 462 maybe ref 78

Line 652 seems to be ref 113, line 713 maybe ref 114, line 767 maybe ref 126

from line 828 is supposed to be a new subsection

from line 1037, Discussion is not a relevant section in this MS, this may be shortened to a Summary section or maybe omitted.

Conclusion is lengthy, please focus.

from line 1184, Abbreviations should be in ABC order

line 1513: ref 137 is not complete

Comments on the Quality of English Language

minor check is necessary

Author Response

Dear Reviewer,

at the very beginning I would like to thank you truly for your support as well as honest and comprehensive feedback. However, I am not sure if I managed with everything I was supposed to.

Q: Author reviews the therapeutic potential of cannabinoids and vit B12. The review is lengthy, not well focused ad not sub-sectioned. It is necessary to revise this review, adding subsections as formatted by „instructions to authors” and „mdpi template”. Please number subsections. It is suggested to shorten the longer subsections , e.g. related to connexins, MS, Vit B12, and also, it is not necessary to add an extra „Discussion„ section.

Author uses as much as 152 references, but many of them do not match the referred text, and still, there are some discrepancies (see specific comments). Thus, the references in the text need an extensive review.

A: I numbered subsections and tried to shorten them properly, however, I enlarged section devoted to vitamin B12 by adding, not only tables (as it also was suggested to me by first Reviewer, but also more comments/descriptions marked by “tracked changes”. I shortened it by reducing references to 126 as well as making corrections including shortening the Conclusion part by removing separate Discussion.

But, of course, I will make further changes if needed.

Q: Please review Title (which seems to be 2 sentences)

(...)

line 207, reference related to TRPA1 is missing line 233, ref related to animal models is missing

(...)

Conclusion is lengthy

from line 1184, Abbreviations should be in ABC order.

A: According to this comment, I a tried to make necessary changes regarding every mentioned aspect. And the manuscript now (without tables and references reduced to 126) has 19 pages. Corrections (despite syntactic - which I made with the help of ChatGPT) are marked by “tracked changes”.

But, of course, I will make further changes if needed.

Round 2

Reviewer 1 Report (Previous Reviewer 2)

Comments and Suggestions for Authors

I consider the article proper to be accepted in the journal. The authors improved it.

Author Response

Dear Reviewer,

Q: I consider the article proper to be accepted in the journal. The authors improved it.

I would like to express my gratitude for your kind words supporting my work and efforts. 

Thank you so much! 

Kind regards,

Anna Kaszyńska

Reviewer 2 Report (Previous Reviewer 1)

Comments and Suggestions for Authors

Still some minor issues have been raised. Please also check carefully all references for their relevancies in the text!

line 107, please indicate refs as [14-18]

line 121, is ref 22 relevant here?

lines 183-187 and 224-235, refs are still missing here!

line 167, please give also author’s name instead of „research”

lines 296-297, please check of refs 49-51 are relevant

lines 297-310, this is confusing, please rephrase

lines 297-303, 478-484, 506-510, 576-, 588-, etc. please omit sentences in brackets and put them into main text. Please rephrase.

line 317, Angelats et al, please give ref number!

lines 625-636, refs for MS are missing

from line 980, ref numbers are missing

lines 980-1003, is that really a Conclusion? (it seems to be rather  a Summary)

Comments on the Quality of English Language

Only minor editing of English is necessary.

Author Response

Dear Reviewer,

First of all, I would like to express my gratitude for your support as well as supervision of my work giving me honest and deep feedback.

Q: line 107, please indicate refs as [14-18]

line 121, is ref 22 relevant here?

lines 183-187 and 224-235, refs are still missing here!

line 167, please give also author’s name instead of „research” lines 296-297, please check of refs 49-51 are relevant

I tried to correct everything that you suggested except correcting two point since (I will attach screenshot to support truthfulness of my words) I have some mistakes in displaying line numbers: I have lack of numbers between 159 and 170, thus I was not always sure if I am referring properly to the issues you mentioned. 

Q: lines 297-303, 478-484, 506-510, 576-, 588-, etc. please omit sentences in brackets and put them into main text. Please rephrase.

line 317, Angelats et al, please give ref number!

lines 625-636, refs for MS are missing

from line 980, ref numbers are missing

lines 980-1003, is that really a Conclusion? (it seems to be rather a Summary)

Moreover, I tried to correct every reference replacing information in brackets with sentences which I mentioned in the whole text as well as Conclusion part and insert reference numbers at the end of the manuscript (or in place you highlighted - however, due to the confusion regarding line numbers, I am not sure if in every places I was supposed to).

Of course, each correction made recently is marked by “track changes”.

Kind regards,

Anna Kaszyńska

This manuscript is a resubmission of an earlier submission. The following is a list of the peer review reports and author responses from that submission.

Round 1

Reviewer 1 Report

Comments and Suggestions for Authors

Author in the manuscript makes a review on the topic of potential therapeutic potential of cannabinoids on neurodegenerative diseases. The topic is interesting, up-to-date and has important clinical relevancies, but this manuscript is not yet ready. The MS has a lot of shortages right now: it has problems of the orientation of sections and subsections, at many sections the discussion is confusing, it does not meet basic formal requirements, the references are not in the proper format as they should be: citations should be numbered in order of their mention in the text, not in Author-Date format. The abstract is lengthy, the formatted text is not according to template and has a lot of missings also at the end of the text, for example Acknowledgments, Conflicts of Interest, etc. Also, lines 713-714 do not meet the standards, it is suggested to omit. Please do an extensive revision on your manuscript following the requirements of the Journal.

Please review sections and subsections. It is suggested to separate Section 1 (Introduction) from others and please put 1/subsections in a separate section (2). Please also omit bold subheadings whithin the text and put them as subheadings into separate lines and start subsections from them, e.g. in lines 151,183,199,206,212,220,225,231,241,258,281, and etc. throughout.

Please introduce all abbreviations at first mention in the text, e.g. THC, CBD, CBG, THCV, etc, also, please omit abbreviations in the headings (e.g. in line 195). Also please conform the stlye of the abbrev, e.g. cannabinoid type 1 receptor (CB1R), arachidonoylglycerol, etc in all places. Also, please use subscripts of the indexes, e.g. CB1, CB2, D1 throughout if its necessary.

Language style of the MS does not meet the requirements of a scientific paper. Please omit the following expressions and rephrase them:

E.g. in Introduction section:

·        this exploratory journey delves into

·        this journey focuses on unraveling the intricate  phenomenon

·        orchestrating a symphony of regulatory responses

·        This intricate dance of molecular events

·        this journey seamlessly extends

·        intricately woven

·        adding another layer to understanding

·        etc.

Specific comments:

line 1: type seems to be a „review”, not a research article

Title is a bit lengthy. Vitamin B12 makes a theoretical disturbance. I would focus it on the therapeutic potential of cannabinoids on neurodegenerative diseases.

Author’s affiliation is missing

Abstract needs a shortening and focusing. CBD is not introduced.

Introduction. Please give a subsection to specify „neurodegenerative diseases”. Please also separate other subsections from introduction into a new section (Section 2). Please omit expressions that do not meet the requirements of scientific papers as mentioned above (e.g. journey, orchestra, intricate dance, etc.).

Lines 63-79, it is not clear what the relationship is of the mentioned mechanisms with cannabinoids.

Section 1.1. and 1.2 has some repetitions, please reorganize these topics (e.g. lines 125-127 and 200-203). It is also suggested to mention main CB agonists in the first section and introduce them. Please correct typewritting of the terms (CB1, CB2, CB1R, arachidonoylglycerol, etc.)

line 137: „neuroinflammation associated with neurodegenerative diseases”, could author specify this term? Each disease related to this term mentioned needs a specification. Please explain main features shortly of the diseases mentioned.

line 151 and throughout the text: „Endocannabinoids Synthesized from Omega Fatty Acids: A Nexus of Lipid Signaling.” These bold subheadings need to put as a subtitle and the text below them into a separate subsection.

Lines 178-182, „In conclusion….” section needs to be put as an end of a section. Please start the next subsection from here in a separate line.

Line 183, „Slower Progression of ALS and Omega-3 Fatty Acids”, if it is a subtitle, please specify ALS. Also please explain main features of the mentioned diseases shortly.

Lines 192-194, it is required to extend this discussion to the synthesizing and degrading enzymes of endocannabinoids.

Line 212, „chronic inflammation, such as neurodegenerative diseases.”, please specify.

Line 227, „misfolded proteins associated with neurodegenerative diseases”, please specify.

Lines 236-240, „In conclusion….” section needs to be put as an end of a section. Please start the next subsection from here in a separate line.

Line 241,” THCV and CBG, „ if it is a subtitle, please specify the abbreviations.

Line 242, CBG, THCV, please specify them at first mention in the text.

Lines 254-256, „While preclinical studies indicate promise, robust clinical trials are imperative to validate the efficacy and  safety of these cannabinoids in neurodegenerative diseases. „, please specify.

Line 258, „Enhancing therapeutic effect by combining with CBD, „ if it is a subtitle, please put it into separate line and specify CBD

Line 267-268, please specify „ neurodegenerative diseases „, „Huntington’s disease „

Line 281, 291, please use subheadings

Line 290, references of the related mechanisms are missing and the exact link with the diseases should be mentioned.

Line 317, „treatment or prevention of neurodegenerative diseases „, please specify

Lines 323-331 section seems to be an Aims of the study, please rephrase to make it clear.

Section 1.4, it is also suggested to use another main section number here.

Lines 335, 363, 403 are also subheadings……

345, 352, 358, why are these underlined?

Line 390, FAAH is not introduced and explained.

Line 392, „ALS and related spectrum disorders like FTD „, please specify

Line 419, „He et al. (2020) and Hill et al. (2019)”, these references do not exist in the ref list!

From line 423, (Khavandi et al., 2022), Kim et al. (2019) , Abate et al., 2021, Cao et al., 2014, (Prinz et al., 2019) and many others are similarly missing in the ref list!

Line 479, references are missing

ine 481, „This journey through the intricate landscape „, please rephrase!

lines 481-497, is it a summary?

Lines 504, 514 are also subtitles?

Line 521, „In the intricate  tapestry of neurodegenerative diseases, „ please rephrase!

Line 537, „the link between the intricacies of protein aggregation”, please rephrase. Please omit „intricate”, „intricacies” words throughout if possible.

Line 540, references should be required.

Line 545, (Xu et al., 2021, Gao et al., 2023) are also not found in the ref list.

Line 569, „offers a nuanced understanding of the mechanisms „, please rephrase

Line 573, „ In conclusion..” may be  put into a new line.

Line 578, Cassano et al., 2017, 2020 is also missing in the ref list.

From line 586, discussion related to vit B12 needs a separate section.

Section 3, please put subheadings into new lines.

Line 625, please specify „principal contributor”

Line 642, (Rakić et al., 2023) is also missing form the ref list.

Line 660, please specify „neurodegenerative diseases”

Line 667, „cannabinoid-based therapeutics”, please specify

Please focus Section 4 (Conclusions)

Line 685, „mitochondrial oxygen consumption (OCR)”, please omit new abbrev in the last section!

References: it is required to format according to Instructions to Authors and put citations in numeric order at their first mention in the text. Since many citations are not found in the ref list and versa, it is suggested to use citation handling program, such as Endnote.

When making careful revision of the text, also please provide an exact „track changes” version to make the reviewer properly to follow the changes of the text made by the Author.

Comments on the Quality of English Language

It needs a moderate revision of English. 

Reviewer 2 Report

Comments and Suggestions for Authors

The document "Unraveling the Therapeutic Potential of Cannabinoids and Vitamin B12: Insights into Modulating the Endocannabinoid System, Mitochondrial Dynamics, and Microglia Function Regarding Aggregation in Neurodegenerative Diseases" provides a comprehensive exploration of the potential therapeutic benefits of cannabinoids and vitamin B12 in addressing neurodegenerative diseases. The document delves into the modulation of the endocannabinoid system, the impact of cannabinoids on protein aggregation and microglial function, and the potential synergies between cannabinoids and vitamin B12. It also discusses the role of cannabinoids in modulating mitochondrial function and cellular communication in the context of neurodegenerative diseases.

While the document discusses the potential therapeutic effects of cannabinoids and vitamin B12, it would benefit from a more in-depth analysis of the limitations and challenges associated with their clinical application. Furthermore, the document could provide a more critical evaluation of the existing literature and research gaps in the field, which would enhance the overall scholarly contribution of the manuscript.

Moreover, the document would benefit from a more explicit discussion of the potential ethical and regulatory considerations associated with the use of cannabinoids and vitamin B12 in clinical settings. This would provide a more comprehensive understanding of the practical implications of the proposed therapeutic strategies.

In conclusion, while the document offers valuable insights into the therapeutic potential of cannabinoids and vitamin B12 in addressing neurodegenerative diseases, it would benefit from a more thorough and critical analysis of the existing literature, research methodology, and practical implications. Addressing these aspects would enhance the scholarly rigor and practical relevance of the manuscript.

Comments on the Quality of English Language

The English quality of the manuscript is good, requiring minor editing

Reviewer 3 Report

Comments and Suggestions for Authors

Anna is reviewing cannabinoids and their functional roles and therapeutic potentials in topics including endocannabinoid system, mitochondrial dynamics, neurodegenerative diseases, etc. The topic is broad and comprehensive, but the author can hardly cover all aspects.  

The author tried also to include Vitamin B12 to further expand the scope of this reviewer. But unfortunately, the function and therapeutic potential of Vitamin B12 is barely discussed on page 12-13. 

Besides, because the topic is too much broad, the author overlooked a good amount of key information. Examples including –

a. Given that the topic involves endocannabinoid system, the discussion should not only be limited to CB1 and CB2. Novel cannabinoid receptors such as GPR18, GPR55, GPR119 etc. deserve to be assessed and summarized. 

b. Cannabinoid receptors are mainstream class A GPCRs. However, the allosteric modulations towards them are largely overlooked in the current review. The allosteric effects of modulators play critical role in the signal transduction.  

c. Regarding signal transduction, beta-arrestin signals, cAMP signals, GTP signals for cannabinoid receptors are ignored in this review. These signals were utilized to form GTPgammaS binding assay, cAMP assay, and b-arrestin assay, which are vastly used in designing synthetic cannabinoid to activate or inhibit cannabinoid receptors. 

d. Etc. 

Another comment to point out is that there is no figure prepared. Figures in review papers are highly desired. They help readers to quickly grasp concepts and ideas the author want to convey, and help summarize the key points in the writing.

Thus, the reviewer would not recommend this review as it only briefly touching as much subdivisions as possible but without depth. The reviewer does not perceive authors opinions nor insights in this review but dry facts. The reviewer would suggest to have the writing better tailored towards a specific scientific question, so that the author would be able to dive deeper into that specific direction.

Comments on the Quality of English Language

Minor editing of English language is required.